# Influence of Photon-Induced Photoacoustic Streaming (PIPS) on Root Canal Disinfection and Post-Operative Pain: A Randomized Clinical Trial

**DOI:** 10.3390/jcm9123915

**Published:** 2020-12-02

**Authors:** Narcisa Mandras, Damiano Pasqualini, Janira Roana, Vivian Tullio, Giuliana Banche, Elena Gianello, Francesca Bonino, Anna Maria Cuffini, Elio Berutti, Mario Alovisi

**Affiliations:** 1Department of Public Health and Pediatrics, University of Turin, 10126 Turin, Italy; janira.roana@unito.it (J.R.); vivian.tullio@unito.it (V.T.); giuliana.banche@unito.it (G.B.); annamaria.cuffini@unito.it (A.M.C.); 2Department of Surgical Sciences, Dental School, University of Turin, 10126 Turin, Italy; damiano.pasqualini@unito.it (D.P.); gianello.elena@gmail.com (E.G.); bonino.francesca@icloud.com (F.B.); elio.berutti@unito.it (E.B.); mario.alovisi@unito.it (M.A.)

**Keywords:** root canal, disinfection, lasers, quality of care, irrigation, pain, post-operative, periodontitis, endodontic therapy

## Abstract

The aim of this study was to evaluate the ability of a PIPS (photon-induced photoacoustic streaming) Er:YAG laser to reduce the root canal system bacterial count in vivo in comparison to the traditional irrigation technique. The post-operative patients’ quality of life (QoL) after endodontic therapy was evaluated through a questionnaire. Fifty-four patients affected by pulp necrosis with or without apical periodontitis biofilm disease were selected for endodontic treatment and randomly assigned to Group A (*n* = 27) with traditional irrigation and Group B (*n* = 27), with PIPS irrigation applied according to the protocol. Shaping was performed with ProGlider and ProTaper Next, and irrigation was performed with 5% NaOCl and 10% EDTA. Intracanal samples for culture tests were collected before and after irrigation. The microbiological analysis was evaluated by the Kolmogorov–Smirnov normality and Mann–Whitney tests (*p* < 0.05). A self-assessment questionnaire was used to evaluate the QoL during the 7 days after treatment; differences were analysed with Student’s *t*-test. Irrigation with the PIPS device was significantly effective in reducing bacterial counts, which were higher for facultative than obligate anaerobic strains, particularly for Gram-negative bacteria, without statistical significance (*p* > 0.05). There were no significant differences among the QoL indicators, except for the maximum pain (*p* = 0.02), eating difficulty (*p* = 0.03) and difficulty performing daily functions (*p* = 0.02) in the first few days post-treatment. PIPS may represent an aid to root canal disinfection not affecting the patients’ QoL, particularly for the first day after treatment.

## 1. Introduction

Apical periodontitis biofilm disease is a periradicular inflammatory disease related to the presence of bacteria and toxins within the root canal system [1,2]. Different studies have focused on the behavior of bacteria and microbial products in the etiopathogenesis of apical periodontitis [2,3]. Several bacteria are responsible for polymicrobial primary endodontic infections: among these, obligate anaerobic bacteria predominate [4]. The obligate anaerobes are rather easily eradicated during root canal treatment. Facultative anaerobes such as non-mutans streptococci (such as *Streptococcus anginosus, S. constellatus, S. intermedius*, enterococci and *Lactobacillus* spp.) are often resistant to chemo-mechanical instrumentation and root canal dressings. In particular, *Enterococcus faecalis* can frequently be isolated from root canals in the case of failed root canal treatments [5]. The only way to achieve periapical healing is endodontic treatment, which aims to reduce the intracanal bacterial load through shaping, cleaning and filling [6]. Previous epidemiological studies showed that the technical quality of the endodontic treatment is a fundamental prognostic factor [7,8]. However, it is impossible to completely shape the root canal walls and isthmuses by using rotary or reciprocating instruments due to the complexity of the endodontic anatomy [9]. Therefore, the cleaning procedure plays a fundamental role in achieving complete disinfection [10]. The current trend of developing less-invasive endodontics, accompanied by the simplification of operational sequences and a reduction in working time, places an even more marked emphasis on this topic [1]. Several irrigant activation methods have been proposed [11,12]. The most commonly used irrigant is sodium hypochlorite (NaOCl). Ethylenediamine tetraacetic acid (EDTA) and citric acid (CA) are the other large group of endodontic irrigants. Chlorhexidine (CHX) is a cationic bisbiguanide antiseptic with broad-spectrum antibacterial activity. The other irrigants used in endodontics include sterile water, hydrogen peroxide, physiological saline, urea peroxide and iodine compounds [11]. Recently, laser technology was proposed to improve the results obtained with traditional disinfection methods [13]. A photon-induced photoacoustic streaming system (PIPS) requires the use of an Er:YAG laser to activate the water molecules contained in the irrigant solutions [14,15,16]. The laser light actively pushes the irrigating solutions three-dimensionally through the isthmuses, anastomoses and lateral canals, allowing a deeper cleaning of the entire root canal system. The extremely low energy level required to activate irrigants is below the dentin ablation threshold due to the very short pulse duration [15,16,17]. However, despite the tip insert position being at the level of the access cavity, there is still a potential risk of the extrusion of irrigants and debris beyond the apex. It is well known that extruded dentinal debris may induce periradicular inflammation, leading to post-operative pain, negatively affecting the patients’ quality of life (QoL) and their subjective assessment of alternative treatments [2,18,19,20]. The primary objective of this randomized clinical trial was to evaluate the ability of the PIPS technique to reduce the intracanal bacterial count in vivo compared to the traditional endodontic irrigation method. The secondary objective was to assess the influence of the PIPS technique on post-operative patients’ QoL.

## 2. Materials and Methods

This randomized controlled clinical trial (two-parallel-group design) was prepared following the CONSORT guidelines [21] and was performed according to the principles of the Helsinki Declaration [22]. This study was authorized by the Local Ethics Committee, and the treatment protocol was explained to all the patients through an information sheet and was approved with informed consent.

Eligibility criteria. Consenting and healthy subjects of both genders were enrolled until the required sample size was reached. The patients presented a first or second maxillary or mandibular molar with a diagnosis of pulp necrosis with or without symptomatic or asymptomatic apical periodontitis biofilm disease. Patients with a sinus tract, periapical abscess or facial cellulitis were not enrolled due to the possibility of confounding the QoL perception regardless of the treatment. Patients with physical or psychological disabilities or an inability to understand the study instructions were excluded. Additionally, patients who had received emergency endodontic care were not included in the study.

Interventions. The medical and dental status and history of each patient were collected. Intraoral examination was performed with 3.5× dental loupes. The pulpal and periradicular statuses were assessed with thermal and electric pulp tests (Diagnostic Unit, Sybron, Orange, CA, USA), palpation and percussion. The periodontal status was also recorded. Preoperative periapical radiographs were obtained using a Rinn XCP (Rinn Corp, ELgin, IL, USA) and a digital Photostimulable Phosphor Plates (PSP) system. The data were processed and archived by a dedicated scanner (OpTime Soredex, Tuusula, Finland). Teeth with a loss of lamina dura and periodontal ligament enlargement >2 mm were classified as having lesions of endodontic origin (LEOs). The clinical and radiological data were analyzed by three blinded Assistant Professors within the Endodontic Department, and their concordance was analyzed according to the Fleiss’ K score until inter-examiner reliability (K > 0.70) was expected. Only teeth with primary root curvatures between 0° and 40° in bucco-palatal or lingual vision were included according to the Schneider method, and teeth with excessive calcification were excluded. The American Association of Endodontists (AAE) difficulty form was filled out for every clinical case [23].

All treatments were performed by a single expert clinician with more than 3 years of experience. After local anesthesia, the teeth were isolated with a rubber dam, and the dental surface was disinfected with a 30% hydrogen peroxide solution. After the removal of the carious tissue, an endodontic pre-treatment with composite was carried out, in order to ensure an adequate irrigation reservoir.

After the access cavity preparation, each canal was irrigated with saline solution, and the pre-shaping microbial sample was taken using sterile paper cones inserted inside the canal up to the point of engagement with the walls (1st sample: baseline bacterial sample). The first file to bind the canal walls was a K-File with a minimum size #15 up to the root canal middle third, in order to standardize the canal width. Afterwards, for all the canals, a manual canal scouting was performed with K-File #10 (Dentsply Maillefer), a mechanical glide path using ProGlider (PG) (Dentsply Maillefer) and mechanical shaping using ProTaper Next (PTN) X1 and X2 (Dentsply Maillefer) up to the working length (WL). Apical patency was established and confirmed with a size #10 K-File 0.5 mm beyond the apex. The electronic WL (EWL) was recorded with an apex locator (Denta Port ZX, J. Morita MFG, Corp. Kyoto, Japan) three times: during canal scouting, with a #10 size stainless-steel K-File; at the end of the glide path, with a #17 size stainless-steel K-File; and during shaping after X1, with a #17 size stainless-steel K-file. At the end of the glide path, a radiographic check of the WL was performed, using a #17 size stainless steel K-File. All the canals in both groups were instrumented to the working length with each instrument, except for the ProTaper Next X2, which was kept 0.5 mm shorter than the working length, to reduce the risk of debris extrusion and to ensure homogeneous study conditions between the traditional irrigation group and the PIPS irrigation group. The apical finishing was then completed with K-Files NiTiFlex #25 at the WL.

Irrigation protocol. Before the second sampling after instrumentation, the patients were randomly assigned to one of the two irrigation groups (Group A or Group B) by a single-blinded operator, and the root canals were irrigated in the two groups with alternating 5% NaOCl and 10% EDTA solutions for a total of 10 mL for each. Standard endodontic manual irrigation (Group A) was performed using a 30 G side-vented steel needle (Kendall, Mansfield, MA, USA) after every instrument before apical finishing. The needle was positioned inside the canal without engagement with the walls and about 3 mm from the working length during the whole process. For the activation of the irrigation solutions, the needle moved with an amplitude of approximately 2 mm. The PIPS group (Group B) was exposed to laser irradiation with an Er:YAG laser (Fotona LightWalker ST-E Ljubljana, Slovenia), characterized by a wavelength of 2940 nm in exposure intervals of 30 s. The laser was set with frequency values of 15 Hz and an energy of 20 mJ, guaranteeing a total power of 0.30 W. The pulse duration was set to the Super Short Pulse (SSP), equal to 50 μs. During use, the air/water spray of the handpiece was deactivated. The control panel had standard PIPS parameters of 20 mJ and 15 Hz in SSP mode (50 μs), resulting in an average power of 0.30 W [17]. A standardized quartz tip (diameter, 600 μm, and length, 9 mm) was used, characterized by having the conical end part uncoated by the polyamide protective sheath. The tip was placed in the coronal part of the pulp chamber, without the need to introduce it into the root canal. During the activation of the laser, a continuous irrigation flow was guaranteed with a plastic syringe with a 30 G side-vented steel needle, positioned inside the canal without engagement with the walls and within 3 mm of the working length. After each instrument, PIPS irrigation for 30 s with 10% EDTA was carried out to clean the canals of the debris. The laser-activated irrigation (LAI) was performed as follows:-Thirty seconds of PIPS with a continuous flow of 10% EDTA (5 mL);-Thirty seconds of PIPS with a continuous flow of sterile distilled water (5 mL);-Thirty seconds of PIPS three times with a continuous flow of 5% NaOCl (5 mL) with 30 s pauses between cycles;-Thirty seconds of PIPS with a continuous flow of sterile distilled water (5 mL) as a final step.

This irrigation protocol was performed after having instrumented the canal with ProTaper Next X2 0.5 mm shorter than the WL, but before the completion of the apical finishing with K-Files NiTiFlex #25, to reduce the risk of the apical extrusion of debris.

For all the groups, after a final profuse irrigation with saline solution, the post-treatment microbial sample was taken with sterile paper cones and inserted within the canal up to 2 mm from the WL (2nd sample). Finally, the canals were re-irrigated with 5% NaOCl and dried with paper cones, and the access cavity was sealed with a temporary filling (IRM, Dentsply International Inc., York, PA, USA). No occlusal adjustments were performed.

Microbiological analysis. Microbial samples collected before (1st sample) and after irrigation treatment (2nd sample) were placed in a sodium thioglycolate broth (Biolife, Milano) and sent to the Bacteriology and Microbiology Laboratory, Department of Public Health and Pediatrics, University of Turin. All the samples were serially diluted 1:10 in normal saline (0.9% NaCl) and plated on selective and differential media to quantify the total microbiota and the number of obligate and facultative anaerobic bacterial strains: in detail, Brain Heart Infusion Agar (BHA, Becton Dickinson, BD, Franklin Lakes, NJ, USA) was used to determine the number of facultative anaerobic strains, Columbia CNA Agar with 5% Sheep Blood (SCNA, BD) was used to determine the total number of Gram-positive anaerobic (obligate and facultative) bacterial strains, and Schaedler Agar with Vitamin K1 and 5% sheep blood (SCV, BD) and Schaedler Kanamycin-Vancomycin Agar with 5% Sheep Blood (SKV, BD) were used for Gram-positive and Gram-negative obligate anaerobes, respectively. All plates were incubated at 37 °C: BHA plates for 24–48 h under aerobic conditions, and the other ones, for about two weeks under anaerobic conditions in an anaerobic system (Gaspak EZ anaerobe pouch system kit, BD). All the aerobic cultures were examined after 24 and 48 h of incubation, whereas the anaerobic cultures were examined for growth every 3 days. The microbial counts are reported as colony-forming units/mL (CFU/mL).

Post-operative pain self-assessment questionnaire. All the patients were scheduled for a subsequent appointment after 7 days to fill the root canals, and they were dismissed with post-operative instructions and a prescription for optional analgesics. The patients’ post-operative pain (mean and maximum) was assessed through a visual analogue scale (VAS) made of a 10 cm line, where 0 = no pain and 10 = unbearable pain. The patients also received a self-assessment questionnaire, in order to evaluate post-operative QoL within the following 7 days after shaping. The following indicators of the patient’s QoL were used: difficulty in chewing, speaking, sleeping, performing daily functions and social relations and overall QoL. A Likert scale (LS) was used with a numerical value from 0 (none) to 10 (very much). In addition, the analgesic intake, evaluated by the number of analgesic tablets taken in the post-operative period, and the number of days to complete pain resolution were also evaluated. The progressively numbered questionnaires were returned anonymously in a collecting box. Only the principal investigator was aware of the correspondence between the codes and patients and was excluded from the data analysis. The root canal obturation was completed in the second appointment after 7 days with the Thermafil^TM^ technique (Dentsply Maillefer) in association with the use of a eugenol-based canal sealer (Pulp Canal Sealer EWT, Kerr Endodontics, Orange, CA, USA). A final post-operative radiograph was performed, and a follow-up was established after 3, 6 and 12 months.

Sample size. To detect a difference of 5% (a change of 0.5 points on the visual scale) in the post-operative pain values between groups, an alpha-error of 0.05 and a power (1-beta) of 80% were considered, requiring a sample size of 23 patients for each group. In the present study, 27 patients for each group were analyzed.

Randomization. The randomized order was obtained using computer-generated tables. The following parameters were considered for randomization to control for potential confounders: the prevalence of pain before treatment, mean pain before treatment and clinical diagnosis. An operator, external from the clinical treatment, prepared blinded envelopes containing the randomized allocation for each patient. The same operator communicated the allocation to the clinician after the initial patient assessment and before root canal instrumentation. The operator was not blinded to the allocation group, as each instrument required a specific technique. However, the randomization, allocation and statistical analysis were performed by blinded operators.

Statistical analysis. The obtained data were analyzed descriptively and inferentially. The randomization, group assignment and statistical analysis were performed by a blinded operator. The distribution of the microbial analysis data was evaluated by means of the Kolmogorov–Smirnov normality test. The differences between the groups were analyzed by the Mann–Whitney test (*p* < 0.05). The variation in the QoL indicators in the 7 post-treatment days was assessed using a specific variance analysis model for repeated measures (two groups compared). Student’s *t*-test was used for the amount of painkillers taken. All the statistical analyses were performed with the SPSS statistical software, version 17.0.

## 3. Results

A total of 80 subjects were selected for inclusion, and 60 patients were finally enrolled and randomized between traditional endodontic irrigation (Group A, *n* = 30) and PIPS irrigation (Group B, *n* = 30). The distributions of occupation, education and socio-economic background between the groups were uniform. Three patients in Group B and two in Group A were lost to follow-up before the second visit, while one patient in Group A required an unscheduled intervention during the observation period, due to a post-operative flare-up. Finally, data analysis was performed on 27 patients for each group (48% male; 52% female; 21% 16–30 years; 37% 31–45 years; 42% 46–60 years). The patient flow and the patients’ baseline characteristics are presented in Figure 1 and in Table 1. Statistical analysis revealed that the demographic and clinical variables were similarly distributed among the experimental groups (*p* > 0.05).

Microbiological analysis. The results obtained from the microbiological analysis are shown in Table 2. A bacterial load reduction (BLR) was observed for both techniques, even if the PIPS laser technology irrigation in Group B showed a greater antibacterial efficacy than the conventional endodontic manual irrigation in Group A for facultative anaerobes (BLR = 98.18%), total anaerobic strains (BLR = 92.6%) and Gram-negative obligate anaerobes (BLR = 100%). These differences were not statistically significant (*p* > 0.05). The correspondence between the BLR and mean and maximum post-operative pain on Day 1 in both groups is reported in Table 3. For each group, the total BLR seemed not to be statistically correlated to the intensity of pain calculated through the VAS scores on Day 1 post-disinfection (*p* > 0.05).

Post-operative pain and quality of life indicators. The baseline QoL indicators did not show statistically significant differences between the groups (*p* > 0.05). Significant results were found between Groups A and B for the following indicators on Day 1 post-operation: maximum pain (*p* = 0.02), eating difficulty (*p* = 0.03) and difficulty performing daily functions (*p* = 0.02) (Figure 2). These indicators appeared to be lower for patients treated with PIPS irrigation. For the other indicators, from Days 1 to 7 post-treatment, the trend values seemed to be lower in the PIPS group but without significance (*p* > 0.05).

## 4. Discussion

The laser energy exploited with the LAI (laser-activated irrigation) methods has been claimed to improve the effectiveness of endodontic irrigants [11,24,25]. The recent PIPS laser technique has the peculiarity of being based on sub-ablative photoacoustic rather than photothermal phenomena, lowering the risk of scaling the dentinal surface [26,27]. The low energy levels (between 20 and 50 mJ), with a frequency between 10 and 15 Hz and with very short pulses (50 ms), generate photoacoustic shockwaves [17]. The shockwaves generated during PIPS activation may create a three-dimensional flow in the root canal system, removing more debris and pulp residues [1,17]. Preliminary in vitro studies reported that the PIPS protocol should remove more microbial biofilm inside the canal system and open more dentinal tubules compared to traditional endodontic irrigation [27,28]. However, no data are available about the efficacy of photoacoustic streaming laser-based irrigation for root canal disinfection in vivo. Moreover, the use of laser-activating techniques may extrude more debris beyond the apical foramen compared to the traditional irrigation methods, negatively affecting the patients’ post-operative pain and QoL [29]. The QoL is a very complex phenomenon related to the state of health and socio-economic and cultural needs in a context of individual standards and expectations, thus concerning physical, psychological and social well-being [30,31]. Patient satisfaction is an important aspect to take into account when performing dental treatment [19,32], and the QoL following dental treatment can be assessed through a self-assessment questionnaire [33,34]. In the present randomized clinical trial, the microbiological analysis was carried out on different types of medium that allow the growth and isolation of different bacterial species within the canals. An in vivo approach is more complex from a microbiological point of view, since the bacterial types are more diversified and there are several species to isolate [2,35]. In order to avoid endodontic sample contamination from carious or oral bacteria, it is important to perform proper tooth isolation with a rubber dam and dental surface disinfection with a hydrogen peroxide solution at 30% [35]. Within the limits of this study, PIPS showed interesting disinfectant efficacy, although this was not statistically different from that of conventional irrigation for facultative anaerobic and Gram-negative obligate anaerobic strains.

However, a correlation between residual bacteria at the end of the instrumentation and the post-operative pain did not emerge. Therefore, it can be assumed that the presence or absence of residual bacteria in the canal is not the direct cause of post-operative pain and worse QoL perception. It is likely that the onset of post-operative pain may be related to the extrusion of debris or irrigant solution beyond the apex [20,36]. Therefore, to standardize the experimental conditions in both patient groups, the canal shaping was made 0.5 mm shorter than the apex, and an apical finishing with K-Fine NiTiflex #25–#30 was ensured. Several literature studies have analyzed the apical extrusion of debris and associated issues [24,37]. The factors influencing this risk are related to the presence of a large, viable apical foramen; the pressure generated in the fluids within the canal when the laser devices are used; the energy applied; and the position of the tip [24,37]. The present study did not report a risk of irrigant solution and debris extrusion beyond the apex due to irrigant activation by the laser, resulting in a better perception of post-operative pain in the PIPS group. The QoL indicators appeared to be more favorable on post-operative Day 1 for the patients enrolled in the PIPS group. As regards the other analyzed variables, the PIPS method appears to generally lead to less discomfort than the traditional irrigation protocol, even if the data are not statistically significant. This appears to contradict the literature data according to which the PIPS presents a risk of irrigant solution and debris extrusion beyond the apex due to the activation and agitation of the irrigant systems by the laser, resulting in major discomfort for the patient [20,36,38]. However, the patients’ post-operative perception is highly subjective, and a larger study population could be advisable.

## 5. Conclusions

Based on the present results, the PIPS method could be considered a viable system for endodontic disinfection, especially in the case of simplified operational protocols and reduced instrumentation times. The improvement of endodontic disinfection systems is a key step for an ever increasingly modern, easy, fast and minimally invasive treatment plan, despite the traditional irrigation protocol, which continues to be extremely competitive with regard to effectiveness in reducing bacterial infections.

## Figures and Tables

**Figure 1 jcm-09-03915-f001:**
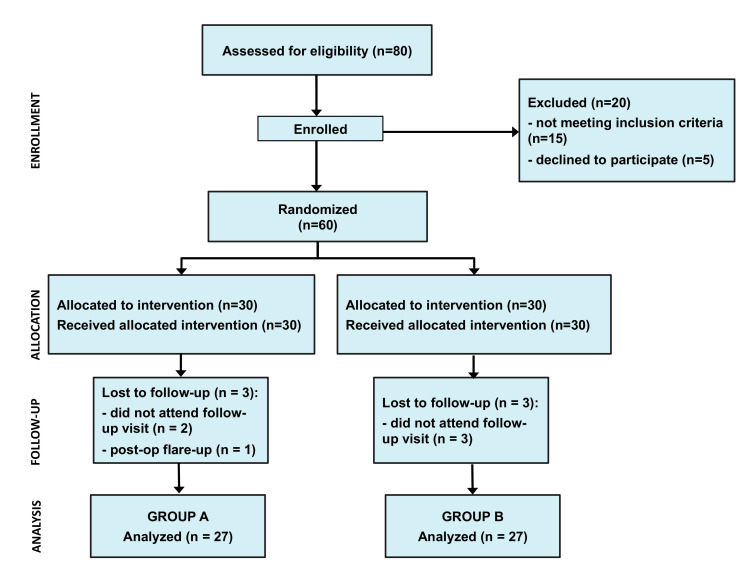
Randomized clinical trial patient flow diagram. Group A: Standard endodontic manual irrigation; Group B: photon-induced photoacoustic streaming (PIPS) laser-activated irrigation.

**Figure 2 jcm-09-03915-f002:**
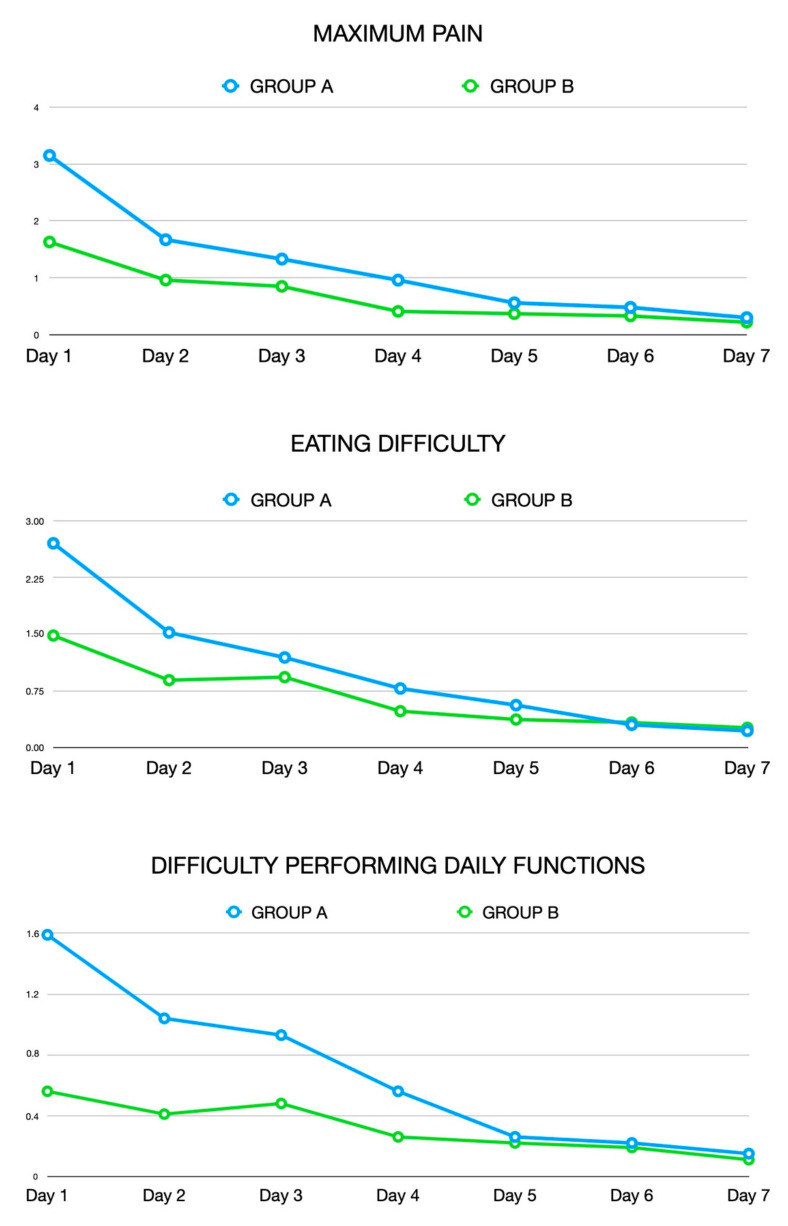
Significant results between Groups A and B for the following indicators on Day 1 post-operation: maximum pain (*p* = 0.02), eating difficulty (*p* = 0.03) and difficulty performing daily functions (*p* = 0.02). Group A, Standard endodontic manual irrigation; Group B, PIPS laser-activated irrigation.

**Table 1 jcm-09-03915-t001:** Group baseline characteristics. Group A, Standard endodontic manual irrigation; Group B, PIPS laser-activated irrigation. AAE = American Association of Endodontists, NS = No statistically significant difference, *p* > 0.05; LEO = Lesion of endodontic origin >2 mm; LS = Likert scale.

Pre-Operative Status	Group A (*n* = 27)	Group B (*n* = 27)	*p* < 0.05
Male	17	14	NS
Female	10	13	NS
AAE difficulty (minimal/moderate/high) (*n*)	11/15/1	19/6/2	NS
Pulp necrosis (%)	100	100	NS
Symptomatic apical periodontitis (%)	51.9	44.4	NS
LEO prevalence (%)	66.7	48.1	NS
Pain prevalence (%)	70.4	63.1	NS
Mean pain score (LS)	4.3	2.9	NS
Maximum pain score (LS)	5.6	3.7	NS
Quality of life (medium)	2.8	1.9	NS

**Table 2 jcm-09-03915-t002:** Bacterial load reduction (BLR) values for different bacterial culture media. Group A, Standard endodontic manual irrigation; Group B, PIPS laser-activated irrigation. Brain Heart Infusion Agar (BHA) to determine the number of facultative anaerobic strains, Columbia CNA Agar with 5% Sheep Blood (SCNA) to determine the total number of anaerobic (obligate and facultative) bacterial strains, and Schaedler Agar with Vitamin K_1_ and 5% Sheep Blood (SCV) and Schaedler Kanamycin-Vancomycin agar with 5% Sheep Blood (SKV) for Gram-positive and Gram-negative obligate anaerobes. SD, Standard deviation.

	Group A (*n* = 27)	Group B (*n* = 27)	*p*-Value
	Mean (%) ± SD	Mean (%) ± SD	
Facultative anaerobes (BHA medium)	94.19 ± 18.5	98.19 ± 3.72	0.95
Facultative and obligate anaerobes (SCNA medium)	89.95 ± 15.1	92.6 ± 15.1	0.37
Gram-positiveobligate anaerobes (SCV medium)	94.83 ± 11.2	91.67 ± 19.7	0.20
Gram-negative obligate anaerobes (SKV medium)	98.43 ± 6.2	100 ± 0.2	0.97

**Table 3 jcm-09-03915-t003:** The correspondence between bacterial load reduction (BLR) and mean and maximum post-operative pain on Day 1. Group A, Standard endodontic manual irrigation; Group B, PIPS laser-activated irrigation.

	Mean Pain (Day 1)			Maximum Pain (Day 1)		
Bacterial Load Reduction (BLR) (%)	Group A (*n* = 27)	Group B (*n* = 27)	*p* < 0.05	Group A (*n* = 27)	Group B (*n* = 27)	*p* < 0.05
100%	2.02	1.72	NS	3.12	1.41	NS
< 100%	2.61	2.12	NS	3.30	1.76	NS

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
