# Peer review of "Influence of Photon-Induced Photoacoustic Streaming (PIPS) on Root Canal Disinfection and Post-Operative Pain: A Randomized Clinical Trial"

_jcm, 2020, doi:10.3390/jcm9123915_

Round 1

Reviewer 1 Report

The present article is entitled as "Influence of Photon-Induced Photoacoustic Streaming (PIPS) on root canal disinfection and post-operative pain: a randomized clinical trial" and it has 2 main goals. The primary objective is to evaluate the ability of the PIPS technique to reduce the intracanal bacterial count in-vivo compared to the traditional endodontic irrigation method. The secondary objective is to assess the incidence of the PIPS technique on post-operative patients’ QoL.

The article is very well described with soundness scientific language and a few aspects that I would like to mention to improve the final version for publication.

1) Nowadays, apical periodontitis is a biofilm disease. Please correct the paper within this information. 

2) Line 117, the sentence is not ok for the article, please rephrase it. 

3) Lines 146, 259, please check "Microbial samples" instead of bacterial samples.

4) Teeth types included in this article? I think it is important to describe to understand better the pain outcomes. Only mandibulars? Molars? 

5) Lines 275-277: please rephrase the sentence meaning, since no statistical differences were obtained. 

Author Response

Dear reviewer, thanks for the comments. We have added what is required

1) Done. Please see the revised version

2) In the revised version the sentence has been changed

3) Done

4) Thank you. First and second maxillary and mandibular molars were selected. This issue has been specified in the materials and methods section

5) We agree. Please see the revised version,  lines 311-313. The sentence has been changed

Reviewer 2 Report

Dear respected author,

Excellent job on the study design, sample selection, and data analysis and interpretation.  The topic is of significance to the field of endodontics and root canal therapy.  This work has proved PIPS to be a viable option for root canal irrigation with the potential for improved canal disinfection and low postoperative pain compared to the needle-based irrigation method.

I have some minor issue that could be addressed: 

Grammatical and sentence structure errors are present throughout the article.

Introduction:

You may want to introduce the reader to the types of irrigation solutions commonly used and their efficacy and limitations

Also, it is worth mentioning the bacterial species that resist current irrigation methods such E. Faecalis.  One of their virulence factors is deep penetration into dentin.

Materials and Methods:

Samples were properly selected, and power analysis was performed with the appropriate statistical methods used. The sample selected has balanced gender distribution and representing age groups.

The Result section was well described.

Discussion:

I highly advise the authors to discuss the limitation of their study at the end of the discussion section.

Kindest Regards, 

Author Response

Dear reviewer, thanks for the comments.

Point 1: Introduction. Thank you. A brief description of the different types of irrigation solutions commonly used in endodontics has been added.

Point 2: Introduction. Please see the revised version page 2, lines 45-49.

Point 3: Discussion. Thank you. The limits of the study have been discussed.

Reviewer 3 Report

Dear Authors

This is an important paper that describes the In Vivo effect of conventional irrigation Vs Pips laser activation .

The paper raises a few questions:

  1. Line 87- Among the analyzed morphologic data there is no mention to the initial width of the root canal therefore no statistical analysis re-width . The initial sampling of bacteria was taken prior to root canal preparation (After the access cavity preparation, each canal was irrigated with saline solution and the pre- shaping microbial sample was taken using sterile paper cones inserted inside the canal up to the point 96 of engagement with the walls (1st sample: baseline bacterial sample). Therefore differences in root canal width may have influenced the bacterial load of the initial sample. If this data is available ( "first file to bind ")  it should be added
  2. Line 100  In the syringe group mechanical preparation with  PTN was done to WL concomitant to irrigation however in the  Pips group mechanical preparation was  done 0.5 mm short of WL  prior to Pips activation to be followed by size 25 file to the WL  w/o further irrigation. though the reason I is clear  it may deprive  the Pips group from thorough disinfection 
  3. line 295 These results might be due to the higher disinfection reached with PIPS disinfection,
    which may predispose for a better postoperative QoL outcome [38 )  The stats.  do not support  this sentence as all the groups were N.S one from each other in all  aspects.
  4. Lines 26  37   some English language editing may be advised 

Author Response

Point 1: Thank you for the comment. To standardize the canal width, the first file to bind the canal walls was set at a minimum size K-File #15 up to the root canal middle third. This issue has been specified in the materials and methods section, please see lines 112-114.

Point 2: Thank you for this comment. In both groups the apical finishing was completed after the irrigation protocol. This issue was better explained in the text: (line 122 and line 132).

Point 3: Thank you. The sentence has been removed.

Point 4: Thank you. The English editing has been completed.